# Regulation of Cell Signaling Pathways and Non-Coding RNAs by Baicalein in Different Cancers

**DOI:** 10.3390/ijms23158377

**Published:** 2022-07-29

**Authors:** Ammad Ahmad Farooqi, Gulnara Kapanova, Sundetgali Kalmakhanov, Gulnur Tanbayeva, Kairat S. Zhakipbekov, Venera S. Rakhmetova, Marat K. Syzdykbayev

**Affiliations:** 1Department of Molecular Oncology, Institute of Biomedical and Genetic Engineering (IBGE), Islamabad 44000, Pakistan; 2Scientific Center of Anti-Infectious Drugs, 75 al-Faraby Ave, Almaty 050040, Kazakhstan; g.kapanova777@gmail.com; 3Al-Farabi Kazakh National University, 71 al-Farabi Ave, Almaty 050040, Kazakhstan; sundetgali.kalmakhanov@gmail.com (S.K.); g_tanbayeva@mail.ru (G.T.); 4Department of Organization and Management and Economics of Pharmacy and Clinical Pharmacy, Asfendiyarov Kazakh National Medical University KazNMU, Tole Bi St. 94, Almaty 050000, Kazakhstan; zhakipbekov.k@kaznmu.kz; 5Department Internal Diseases, Astana Medical University, Nur-Sultan 010000, Kazakhstan; venerarakhmetova@gmail.com; 6Department of Anesthesiology, Reanimatology and Narcology, Semey Medical University, Semey 071400, Kazakhstan; marat.syzdykbayev@nao-mus.kz

**Keywords:** cancer, metastasis, cell signaling, natural products, baicalein

## Abstract

Landmark discoveries in molecular oncology have provided a wide-angle overview of the heterogenous and therapeutically challenging nature of cancer. The power of modern ‘omics’ technologies has enabled researchers to deeply and comprehensively characterize molecular mechanisms underlying cellular functions. Interestingly, high-throughput technologies have opened new horizons for the design and scientific fool-proof evaluation of the pharmacological properties of targeted chemical compounds to tactfully control the activities of the oncogenic protein networks. Groundbreaking discoveries have galvanized the expansion of the repertoire of available pharmacopoeia to therapeutically target a myriad of deregulated oncogenic pathways. Natural product research has undergone substantial broadening, and many of the drugs which constitute the backbone of modern pharmaceuticals have been derived from the natural cornucopia. Baicalein has gradually gained attention because of its unique ability to target different oncogenic signal transduction cascades in various cancers. We have partitioned this review into different sub-sections to provide a broader snapshot of the oncogenic pathways regulated by baicalein. In this review, we summarize baicalein-mediated targeting of WNT/β-catenin, AKT/mTOR, JAK/STAT, MAPK, and NOTCH pathways. We also critically analyze how baicalein regulates non-coding RNAs (microRNAs and long non-coding RNAs) in different cancers. Finally, we conceptually interpret baicalein-mediated inhibition of primary and secondary growths in xenografted mice.

## 1. Introduction 

Based on the insights gleaned from single-gene to genome-wide scales, it is becoming apparent that cancer is a heterogeneous disease. What emerges is a highly complicated and orchestrated network of receptors that form higher-order ligand–receptor complexes for intracellular transmission of the signals [1,2]. Nonetheless, its mechanism of action remained elusive, until a series of breakthroughs in the 1980s and 1990s cracked open the mystery of extra-ordinary mechanistic insights related to a myriad of oncogenic signaling cascades in different cancers [3,4].

Natural products are rich sources of medicinally and pharmacologically important bioactive chemicals [5,6,7,8]. Baicalein has a molecular weight of 270.24 g/mol. Baicalein has been shown to improve disease severity by regulation of different disease-related proteins [9,10,11,12,13,14,15,16]. Therefore, before a comprehensive and mechanistic analysis of baicalein-mediated cancer chemopreventive properties, we present an overview of antibacterial, antiviral, and oxidative-stress-modulating activities of baicalein.

***Antibacterial Properties****:* The NorA efflux protein is frequently expressed in multidrug-resistant (MDR) *Staphylococcus aureus* strains and is characterized as the main experimental model in the exploration for agents that efficiently inhibit active efflux mechanisms. Baicalein caused a significant reversal of the ciprofloxacin resistance of MRSA by inhibition of the NorA efflux pump [17].

Biofilms formed by *Staphylococcus aureus* remarkably enhanced resistance against antibiotics by hindering the penetration of antibiotics. Importantly, baicalein inhibited *Staphylococcus-aureus*-induced formation of the biofilms, destroyed biofilms, increased vancomycin permeability, suppressed the production rates of staphylococcal enterotoxin A and α-hemolysin, and inhibited quorum sensing systems [18].

Baicalein reduced the *Pseudomonas-aeruginosa*-induced secretion of the inflammatory cytokines, particularly interleukin-1β, interleukin-6, interleukin-8, and TNFα. Furthermore, baicalein suppressed *Pseudomonas-aeruginosa*-induced activation of the MAPK and NF-κB-driven transduction cascades in co-cultured macrophages [19].

The CTX-M-type β-lactamases confer resistance to expanded-spectrum cephalosporins. Baicalein and cefotaxime synergistically reduced the expression of CTX-M-1 [20].

Serotype 2 (SS2) of *Streptococcus suis* has been noted to cause severe health complications. Baicalein and ampicillin combinatorially reduced inflammation and pathological damages such as high infiltration rates of inflammatory cells, alveolar interstitial congestion, and edema in the brain and lung of mice intraperitoneally infected with SC19 [21]. There are direct pieces of evidence which highlight nanoparticle-mediated delivery of baicalein for effective antibacterial activity [22,23].

***Antiviral Properties****:* Halogenated baicalein was found to be a promising antiviral agent against SARS-CoV-2 main protease [24]. Baicalein and gallocatechin gallate inhibited the activities of SARS-CoV-2 main protease and blocked replication of the virus. COVID-19 and sepsis have the ability to pathologically trigger cytokine storms. Levels of serum inflammatory factors interleukin-1α, TNFα, interleukin-4, and interleukin-10 in the model group were increased significantly, while they were reduced considerably in mice treated with gallocatechin gallate and baicalein [25]. Baicalein inhibited SARS-CoV-2 RNA-dependent RNA polymerase and exhibited significant antiviral activities [26].

***Oxidative stress****:* Intracerebral hemorrhage (ICH) is a critical and life-threatening subtype of stroke. ICH animal models were established by injection of collagenase into the right basal ganglia. Baicalein led to an increase in the levels of serum SOD and GSH-Px, whereas neuronal apoptosis and pathological injuries of the brain tissues were greatly mitigated. microR-106a-5p directly targeted PHLPP2 but overexpression of PHLPP2 caused reversal of baicalein-mediated effects on ICH mice. Baicalein activated the NRF2/ARE pathway by suppression of PHLPP2 expression [27].

Baicalein enhanced cellular antioxidant defensive capacities through significant reduction in the levels of ROS generation and the activation of the NRF2 transduction cascade, thus protecting C6 cells from H_2_O_2_-induced damage of neurons [28]. Co-treatment with H_2_O_2_ and baicalein completely suppressed the activation of the apoptotic pathway by upregulation of NRF2 expression and reduction in the levels of ROS [29].

Baicalein led to the protection of cardiomyocytes against oxidative-stress-mediated injuries through the NRF2/KEAP1 cascade. Baicalein effectively induced disassembly of NRF2 and KEAP1. Consequently, NRF2 moved from the cytoplasm to the nucleus and stimulated NRF2/heme oxygenase-1 contents [30].

Baicalein improved the mortality rates, degeneration of neurons, brain water contents, and cerebral vasospasm in rat models of subarachnoid hemorrhage repeatedly injected with autologous blood. Baicalein also switched on the antioxidant mechanism by activating the functions of SOD and catalase and decreased the levels of malondialdehyde [31].

Baicalein ameliorated myocardial ischemia through reduction in oxidative stress and inflammation [32].

Baicalein improved brain injuries after intracerebral hemorrhage by inhibition of ROS-NLRP3 inflammasomes [33]. Overall, baicalein has been demonstrated to improve disease-associated pathological conditions [34,35,36].

After a brief description of additional medicinally significant properties of baicalein, we now exclusively concentrate on the cancer chemopreventive features of baicalein. There are some good reviews related to the cancer chemopreventive role of baicalein [37,38,39,40,41], however, we extensively analyze baicalein-mediated regulation of oncogenic signaling pathways. For the framework of the mini-review, we extensively browsed PubMed using diverse keywords, particularly “baicalein and cancer”, “baicalein and metastasis”, and “baicalein and signaling”.

In this mini-review, we summarize the regulation of WNT/β-catenin, AKT/mTOR, matrix metalloproteinases, JAK/STAT, MAPK, and NOTCH pathways by baicalein in different cancers. We also highlight the existing knowledge gaps in our understanding related to recently available evidence regarding the role of baicalein in the regulation of non-coding RNAs. Finally, we discuss pressing questions and important concepts that should be the focus of future research related to the clinical translation of the pharmacological properties of baicalein.

## 2. Regulation of WNT/β-Catenin Pathway

In the absence of signals, β-catenin becomes marked for degradation by a multi-component “destruction complex” consisting of the scaffolding molecules axin, adenomatous polyposis coli (APC), casein kinase (CK), glycogen synthase kinase-3 (GSK3), and SCF (SKP1/Cullin/F-box)-β-TrCP E3-ubiquitin ligase [42,43]. Importantly, APC and axin recruited β-catenin, and after sequential phosphorylations of specified amino acid residues in the amino-terminal region, β-catenin is poly-ubiquitinated by β-TrCP and tagged for degradation by the proteasomal machinery. Wnt signaling disrupted the functions of destruction complexes, consequently resulting in the stabilization and transportation of β-catenin to the nucleus for transcriptional regulation of target gene networks [44,45].

With the breakneck speed of advancements in various dimensions of evidence-based medicine and modern pharmacology, different approaches for pharmaceutical targeting of the Wnt/β-catenin cascade in cancers have already started the journey on the road to clinical trials. Importantly, experimentally verified evidence has proven the benefits of targeting of Wnt/β-catenin signaling in various human cancers, but it is still worthwhile to further dissect the true potential of baicalein as a safe inhibitor of the Wnt/β-catenin signaling pathway against a wide variety of cancers.

In this section, we summarize how baicalein-mediated targeting of WNT/β-catenin inhibited cancer progression.

β-catenin transcriptionally repressed FOXA2. FOXA2 stimulated the expression of lncRNA-NEF. LncRNA-NEF interacted physically with β-catenin and increased the binding of GSK3β with β-catenin and facilitated phosphorylation and degradation of β-catenin by proteasomal machinery (Figure 1) [46].

Baicalein not only reduced β-catenin but also restricted its nuclear accumulation. The lncRNA-NEF level was found to be enhanced in baicalein-treated osteosarcoma cells. LncRNA-NEF also inactivated the WNT/β-catenin cascade. Intraperitoneal injections of baicalein hampered the progression of osteosarcoma in mice injected with MG63 cells into the medullary cavities of the right tibia. Baicalein exerted tumor suppressive effects by reducing the tumor burden in orthotopic intra-tibia tumor-bearing models [47].

SNHG1 antagonized miR-3127-5p-mediated targeting of FZD4 (Figure 1). Ectopic expression of SNHG1 remarkably increased Wnt/β-catenin signaling but inhibition of SNHG1 remarkably blocked Wnt/β-catenin target gene networks. Ectopic expression of SNHG1 antagonized baicalein-mediated inhibitory effects on Wnt/β-catenin signaling. Baicalein significantly impaired the growth of subcutaneous xenografts in mice inoculated with SNHG1-overexpressing HeLa cells [48].

GSK-3β-mediated phosphorylation makes β-catenin a recognizable target for ubiquitylation and consequent proteasomal degradation, thus inhibiting nuclear accumulation of β-catenin (Figure 1). Baicalein stimulated miR-25 expression and reduced β-catenin levels. Furthermore, GSK-3β level was reported to be enhanced in baicalein-treated cancer cells [49].

Baicalein and docetaxel combinatorially promoted degradation of β-catenin and inhibited nuclear import of β-catenin. Baicalein and docetaxel synergistically reduced tumor burden and tumor angiogenesis in mice subcutaneously xenografted with A549 cells [50].

CCND1 potentiated nuclear transportation of β-catenin and promoted the binding of β-catenin to promoter regions of NANOG, MMP2, and SOX2. Baicalein efficiently inhibited the loading of β-catenin to promoters of NANOG, MMP2, and SOX2. Baicalein suppressed proliferation potential induced by overexpression of CCND1 in HeLa cells [51].

1,2-dimethylhydrazine dihydrochloride (DMH)- and dextran sodium sulfate (DSS)-treated animals supplemented with ruthenium baicalein complexes remarkably enhanced the expression of Bax, whereas expression levels of WNT and β-catenin were found to be profoundly reduced [52].

WNT/β-catenin signals are transduced intracellularly, and β-catenin moves into the nucleus and triggers TCF/LEF transcriptional factors, thus stimulating the transcription of target genes, including c-myc, cyclin D1, and survivin [53]. Importantly, c-myc, cyclin D1, survivin, and β-catenin were evidently suppressed in baicalein-treated 143B and MG-63 osteosarcoma cells. Baicalein-mediated cancer-inhibitory effects were impaired in osteosarcoma cells which exogenously expressed β-catenin. However, baicalein-induced apoptotic effects were noted to be enhanced in β-catenin-silenced cells. Baicalein evidently impaired the tumor growth rates of 143B xenografts in mice models [53].

Baicalein inhibited the proliferation capacities of Jurkat cells by downregulation of β-catenin and its downstream targets [54].

Baicalein increased the apoptotic death of MG-63 cells by increasing the levels of axin, adenomatous polyposis coli, GSK-3β, and CK. There was a notable decline in the levels of β-catenin and c-Myc [55].

Overall, these clues of evidence inform us about the potent role of baicalein as a WNT/β-catenin pathway inhibitor. Hopefully, better studies related to anticancer and metastasis-inhibitory roles of baicalein in animal model studies will galvanize the translatability of this product as a significant clinical agent.

In the upcoming section, we analyze how baicalein inhibited the AKT/mTOR pathway for cancer inhibition.

## 3. Regulation of AKT/mTOR Pathway

The identification and development of small-molecule inhibitors of the pathway have led to the acceleration of the availability of comprehensive crystal structures for complex protein assemblies, the generation of synthetic drug-like compound libraries, and the collection of natural products for high-throughput screening. AKT phosphorylates TSC2 (tuberous sclerosis complex 2) on different sites for the inhibition of the activity of GAP (GTPase-activating protein) for RHEB (RAS homologue enriched in brain). Importantly, GTP-loaded RHEB activates mTORC1 (mammalian TOR complex 1) [56,57]. However, TSC1/TSC2-mediated hydrolysis of GTP to GDP switched RHEB from a functionally active GTP-bound state to an inactive GDP-bound state and inhibited the functions of mTOR. In addition, 4E-BP1 has a major role in translational repression. It inhibits cap-dependent translation and subsequent assembly of the translation initiation complexes. Dissociation of 4E-BP1 from eIF4E depends on the step-by-step phosphorylation of critical amino acids by mTORC1 [58,59].

This section principally deals with the regulation of the AKT/mTOR cascade by baicalein and how inactivation of this pathway leads to the inhibition of carcinogenesis and metastasis.

The migratory and invasive potential of PC-3 and DU145 cells was inhibited by baicalein. Treatment of DU145 and PC-3 cells with baicalein resulted in a marked suppression in the levels of p-AKT and p-mTOR [60].

Levels of mTOR were significantly reduced by the combinatorial treatments with baicalein and docetaxel. Both drugs also inhibited activation of AKT in anaplastic thyroid cancer 8505c cells [61]. Collectively, baicalein reduced the invasive and metastasizing potential of thyroid cancer cells.

mTORC1 inhibitors induced the expression of CD133 in cancer cells, enriched CD133+ cancer cells, and limited the efficacy of anti-cancer drugs. Temsirolimus (CCI-779), a novel inhibitor of mTOR, induced CD133 in tumor-initiating stem-cell-like cells (TICs) as well as Huh7 cells. Nevertheless, co-treatments with baicalein completely prevented this induction and synergistically enhanced cytotoxicity in TICs. SAR1B knockdown sensitized TICs to CCI-779 cytotoxicity. SAR1B loss phenocopied baicalein-mediated cytotoxic effects on CCI-779-treated TICs. NSGTM mice transplanted with HCC tissues resected from a NASH patient significantly prevented tumor growth by temsirolimus and baicalein [62].

Total flavonoid aglycones extracts (TFAE) obtained from *Scutellaria baicalensis* efficiently inhibited tumor growth. TFAE was reconstituted with increasing concentrations of baicalein, wogonin, and oroxylin-A. Importantly, reconstructed TFAE (reTFAE) in combination with TWIST silencing caused a marked increase in E-cadherin and simultaneous reduction in the levels of the PI3K/AKT pathway and N-cadherin [63].

Levels of p-AKT, p-mTOR, NF-κB, and p-IκB were found to be downregulated significantly in baicalein-treated MDA-MB-231 and MCF7 cells. Baicalein remarkably lowered the level of p-AKT in tumor tissues of mice subcutaneously implanted with either MCF7 or MDA-MB-231 cancer cells [64].

Baicalein in combination with cisplatin effectively reduced p-AKT and p-mTOR in drug-resistant SGC-7901 cancer cells (Figure 2) [65]. Additionally, baicalein-mediated inactivation of the AKT/mTOR pathway has also been demonstrated in cervical cancer cells. Baicalein inactivated AKT and mTOR in cervical cancer cells through upregulation of circular RNA (circHIAT1) and inhibition of miR-19a-3p [66]. Importantly, baicalein induced shrinkage of the tumor mass in mice subcutaneously inoculated with circHIAT1-expressing HeLa cells. However, baicalein-mediated tumor inhibition was partially reduced in mice inoculated with circHIAT1-silenced HeLa cells [66].

Baicalein reduced protein levels and phosphorylated forms of mTOR, p70S6K, and 4EBP1 in HeLa cells [67].

TSC1/TSC2 heterodimers, through a critical GAP (GTPase-activating protein) domain located in TSC2, facilitated the hydrolysis of GTP to GDP and switched RHEB from an active GTP-bound form to an inactive GDP-bound form and inhibited the functions of mTOR. DDIT4 has been documented to block the activity of mTORC1 by activation of TSC1/2 complexes. Inactivation of mTORC1 reduced the activation of downstream proteins, including ribosomal S6 protein kinases (S6K1 and S6K2). These kinases classically phosphorylated their downstream ribosomal S6 proteins. However, mTORC1 inactivation led to inhibition of ribosomal S6 protein kinases and S6. DDIT4-mediated inactivation of mTORC1 resulted in the repression of phosphorylation of S6K1 and downstream S6 (Figure 2). Intraperitoneal injections of baicalein induced tumor shrinkage in SCID-Bg mice orthotopically implanted with MDA468 cancer cells. Moreover, levels of DDIT4 were found to be enhanced in the tumor tissues [68].

Baicalein-mediated activation of AMPK switched on ULK1 and reduced the inhibitory effects of mTOR on ULK1. mTOR interacted with ULK1 through RAPTOR and inactivated ULK1 through phosphorylation at serine-757. Baicalein reduced the protein levels of mTOR and RAPTOR in PC-3 cells [69]. Collectively, baicalein acted as a novel autophagy inducer and operated through the activation of the AMPK/ULK1 pathway and inhibition of mTORC1.

## 4. Regulation of Matrix Metalloproteinases

Baicalein considerably reduced the motility and invasiveness of B16F10 cells. Baicalein reduced the activity and levels of MMP2 and MMP9. However, levels of tissue inhibitor of metalloproteinase-1 and -2 were found to be increased concomitantly [70].

Baicalein significantly inhibited nuclear accumulation of NF-κB. Additionally, baicalein also significantly suppressed the phosphorylation of IκBα. The p38-MAPK signaling pathway is involved in NF-κB activation in ovarian cancer cells. Baicalein inhibited MMP2 expression and cell invasion by inhibition of the p38-MAPK-dependent pathway [71].

Baicalein inhibited TNF-α-induced activation of NF-κB and transcriptional regulation of NF-κB-regulated target genes [72].

Flavonoid chemicals in *Scutellaria baicalensis* effectively inhibited nicotine-induced lung-cancer-associated inflammation and metastasis. Baicalein, baicalin, and wogonin reduced MMP2 and MMP9 levels [73].

Baicalein dramatically reduced the phosphorylated forms of MEK1 and ERK1/2. Furthermore, overexpression of MEK1 led to a partial blockade of the metastasis-inhibitory effects of baicalein. Moreover, combinatorial treatments with baicalein and ERK inhibitors synergistically reduced MMP2, MMP9, and uPA and induced an increase in TIMP1 and TIMP2 [74].

Baicalein dose-dependently suppressed the levels of MMP2 and MMP9 in glioma cells [75]. Baicalein-mediated inhibitory effects on MMP2 and MMP9 have also been reported in bladder cancer, hepatoma, and colorectal cancer [76,77,78].

Baicalein inhibited benzo(a)pyrene [B(a)P]-induced pulmonary carcinogenesis in rodent models and reduced the levels of MMP2 and MMP9 [79].

## 5. Regulation of JAK/STAT Pathway

The STAT family of proteins is involved in signal transduction and regulation of transcription. Janus kinase (JAK)-mediated phosphorylation of STAT proteins induced activation of the signaling cascade [80,81,82]. In this section, we exclusively analyze how STAT3 promoted carcinogenesis and how STA3 can be pharmaceutically exploited for cancer inhibition.

Baicalin and baicalein-mediated suppression of PD-L1 was impaired in STAT3-overexpressing SMMC-7721 and HepG2 cells. The number of CD8+ cells was apparently higher in the tumor tissues from baicalin and baicalein-treated BALB/c mice. Furthermore, PD-L1 expression was significantly downregulated in the tumor tissues of the baicalin and baicalein-treated xenografted mice [83].

Baicalein interfered with the activation of STAT3 in 4T1 cancer cells. Baicalein considerably hampered the formation of pulmonary metastatic nodules in mice injected with 4T1 cancer cells (Figure 3) [84].

Baicalein not only inhibited IL-6-mediated phosphorylation of JAK, STAT3, and AKT, but also repressed STAT3-driven upregulation of BCL-XL in multiple myeloma cells (Figure 3) [85].

miR-183 knockdown significantly reduced apoptotic cell death. Baicalein promoted the radiosensitivity of Hela cells via inhibition of the JAK2/STAT3 signaling cascade [86]. Collectively, the use of tumor suppressor miRNA mimics will enhance the efficacy of baicalein.

## 6. Regulation of MAPKs

Inhibition of MEK/ERK1/2 interfered with baicalein-triggered AMPKα phosphorylation. Inhibition of MEK/ERK1/2 and AMPK abrogated baicalein-induced increase in the levels of FOXO3a and RUNX3. RUNX3 knockdown led to significant attenuation of baicalein-induced protein expression of FOXO3a. Baicalein enhanced apoptosis in RUNX3- and FOXO3a-expressing cancer cells. However, AMPK inhibition blocked baicalein-induced caspase 3/7 activity [87].

Baicalein impaired growth rates of the tumor in BALB/c-nude mice inoculated subcutaneously with MG-63 cells. Importantly, p-ERK levels were found to be reduced in tumor tissues of xenografted rodent models [88].

cAMP-mediated PKA activation regulated VASP (vasodilator-stimulated phosphoprotein) phosphorylation in platelets. VASP phosphorylation is associated with attenuation of integrin αIIbβ3-driven downstream signaling and aggregation of the platelets. Integrin αIIbβ3 binds to different arginine-glycine-aspartic-acid containing ligands, including von Willebrand factor, fibronectin, fibrinogen, and fibrin. Pre-treatment with baicalein induced phosphorylation of VASP. Agonist-mediated ERK2 and p38-MAPK phosphorylations were suppressed remarkably by baicalein pre-treatments. Furthermore, baicalein potently inhibited phosphorylation of AKT. Importantly, rat C6 glioma cells induced full activation of the platelets. However, C6-cell-induced platelet aggregation was impaired by baicalein. Baicalein also efficiently prevented platelet/tumor cell interactions [89].

## 7. Regulation of NOTCH Pathway

The NOTCH pathway has been reported to be centrally involved in cancer progression. Baicalein-mediated targeting of the NOTCH pathway resulted in the inhibition of carcinogenesis.

Baicalein suppressed the proliferation of cervical cancer cells via NOTCH-1 and its target genes HES-1 and HES-5 [90].

Baicalein downregulated NOTCH-1 and HES-1 in H1299 and A549 cells, which indicated that baicalein suppressed the NOTCH signaling pathway [91].

Existing evidence is preliminary and future studies must converge on the testing of baicalein in xenografted mice for better analysis of baicalein-mediated inhibitory effects on the NOTCH pathway.

## 8. Regulation of Non-Coding RNAs

Discovery of non-coding RNAs (ncRNAs) has caused a paradigm shift in our understanding regarding the regulation of protein networks and cell signaling pathways in different cancers. Experimental verifications and validations related to microRNAs, long non-coding RNAs, and circular RNAs have led to exciting advancements in various facets of molecular oncology [92,93,94,95,96,97,98,99,100,101].

Ezrin is a target gene of miR-183 (Figure 4). There was an evident increase in the expression of miR-183 and simultaneous suppression of ezrin in baicalein-treated Saos-2 and MG-63 cells. Importantly, ezrin overexpression and miR-183 inhibition abolished baicalein-mediated inhibitory effects on migration and invasion of Saos-2 and MG-63 cells [102].

Both baicalein treatments and overexpression of miR-3663-3p led to the downregulation of SH3GL1 and inactivation of ERK1/2, EGFR, and NF-κB/p65 transduction cascades. Tumors derived from miR-3663-3p-overexpressing Bel-7404 and SK-Hep-1 cells were smaller in size. Furthermore, levels of p-NF-κB/p65, p-ERK, and p-EGFR were reported to be profoundly reduced in the tumor tissues. Importantly, intraperitoneal injections of baicalein induced tumor retrogression in mice subcutaneously xenografted with Bel-7404 or SK-Hep-1 cells [103].

PAX8-AS1-N antagonized miR-17-5p-mediated targeting of PTEN, ZBTB4 (Zinc finger and BTB domain containing 4) and CDKN1A (Figure 4). PAX8-AS1-N knockdown promoted growth of breast cancer xenografts and baicalein-mediated growth inhibition was attenuated significantly by PAX8-AS1-N knockdown [104].

Baicalein downregulated miR-424-3p, upregulated PTEN and reduced the levels of PI3K and p-AKT in H460 and A549 cells. PTEN is a tumor suppressor and inactivates PI3K/AKT-driven signaling. PTEN has been reported to be directly targeted by miR-424-3p. It was shown that miR-424-3p overexpression or PTEN silencing partially weakened baicalein-mediated repressive effects on H460 and A549 cells [105].

BDLNR (baicalein downregulated long non-coding RNA) physically associated with YBX1 and promoted its binding to the PIK3CA promoter and activated PIK3CA expression and the PI3K/AKT pathway (Figure 4). Baicalein-mediated tumor suppression was significantly impaired in mice inoculated with BDLNR-overexpressing HeLa ells [106].

Keeping in view the fact that miRNAs and lncRNAs drive malignant phenotypes from multiple perspectives, in this section, we focus on baicalein-mediated effects on critical signaling cascades modulated by miRNAs and lncRNAs in cancers to demonstrate an up-to-date understanding of this area of research.

## 9. Animal Models

HIF-1α is rapidly degraded by prolyl hydroxylase domain proteins (PHDs). These are 2-oxoglutarate/iron-dependent dioxygenases and utilize molecular oxygen for hydroxylation of specified prolyl residues present within oxygen-dependent degradation domains of α subunits. Insights gained from the functional analysis of von Hippel-Lindau proteins (pVHL) have provided important information about regulation of client proteins. Essentially, hydroxylated subunits of HIF-α are identified by pVHL for subsequent poly-ubiquitination and degradation by proteasomal machineries. Baicalein caused a reduction in HIF-1α levels by promoting its interactions with PHD2 and pVHL, enhanced ubiquitin ligase-mediated degradation by proteasomal machinery, inhibition of nuclear translocation, binding to the hypoxia-response elements, and transcriptional activities of HIF-1α (Figure 5). Additionally, baicalein enhanced tamoxifen sensitivity in MCF-7TR-derived xenografts in NOD/SCID mice. Tamoxifen and baicalein combinatorially reduced the levels of HIF-1α in tumor tissues [107].

During amino acid sufficiency, MAP4K3 is phosphorylated at serine-170 and activates mTORC1. However, during restriction of amino acids, MAP4K3 interacts with PP2A and undergoes dephosphorylation at serine-170, leading to inhibition of MAP4K3 and inactivation of the mTORC1 signaling pathway. MAP4K3 is pivotal for phosphorylation of mTORC1 downstream signaling proteins such as S6K1. MAP4K3 overexpression promoted activation of S6K1 but MAP4K3 knockdown caused a reduction in size of the cells. MAP4K3 interacted with and phosphorylated TFEB (transcription factor EB) and caused cytoplasmic retention of TFEB, leading to autophagy inhibition. Baicalein effectively enhanced proteasomal degradation of MAP4K3 in H1299 and A549 cancer cells. Baicalein-induced degradation of MAP4K3 promoted nuclear translocation of TFEB. Phosphorylation of TFEB facilitated the association between TFEB and 14-3-3 proteins, thereby inhibiting nuclear accumulation of TFEB. Baicalein caused disassembly of TFEB/14-3-3 complexes and promoted nuclear translocation of TFEB. MAP4K3 knockdown sufficiently mimicked baicalein-induced autophagy. There was a notable shrinkage of H1299 xenografts in animal models. Moreover, MAP4K3 expression was found to be reduced in tumor tissues of baicalein-treated xenografted animal models [108].

High glucose concentrations promoted invasion of HepG2 cells. There was a significant increase in HepG2 tumors in the diabetic rodent model as compared to the normal mice. Essentially, HKDC1 antibodies and baicalin limited the rapid development of tumors derived from HepG2 cells in diabetic mice. m6A of HKDC1 facilitated rapid development of liver cancer induced by type 2 diabetes. The m6A writer complex contains the core METTL3 (methyltransferase-like protein 3) and its adaptors. This multi-component machinery is localized in the nucleus, and it adds m6A to mRNAs. Baicalin suppressed type 2 diabetes and liver cancer development by reduction in the m6A level of HKDC1 by repressing the levels of m6A-related gene, METTL3. There was a significant reduction in the m6A (2854 site) levels of HKDC1 in METTL3-silenced cells [109].

SHH, SMO, and GLI-2 were downregulated in baicalein-treated pancreatic cancer cells. Baicalein impaired tumor growth in BALB/c nude mice inoculated subcutaneously with PANC-1 stem cells. Immunohistochemistry analysis indicated that tumor tissues demonstrated significant downregulation of stem cell markers and lowered the levels of SHH pathway receptors and effectors [110].

Pharmacologic targeting of HDACs induced differentiation, inhibited proliferation, and induced apoptotic death in AML cells. Blockade of baicalein-induced degradation of HDAC1 by proteasome inhibitors indicated that baicalein-induced HDAC1 degradation by baicalein was dependent on the ubiquitin proteasome pathway. HSP90 (heat shock protein-90) stabilized client oncogenic proteins. AML1-ETO, an oncogenic fusion protein, recruited transcriptional repressor complexes including HDAC1 to repress AML1-regulated target genes (Figure 5). Interactions between HSP90 and AML1-ETO were disrupted by baicalein-induced acetylation on lysine residues of HSP90. Baicalein caused significant reduction in the amount of huCD45+ cells in spleen and bone marrow of NOD/SCID mice injected with either Kasumi-1 or ME-1 cells [111].

Treatment with baicalein considerably increased the rate of survival of U87-inoculated mice. Moreover, baicalein caused significant reduction in the water content of tumors and ipsilateral cerebrums. Baicalein effectively suppressed the uptake of sucrose by U87 gliomas and findings provided clues that baicalein significantly reduced permeability of U87 gliomas. The HIF1α/VEGF pathway played an important role in the edema induced by gliomas. Treatment of tumor-bearing mice with baicalein suppressed HIF1α, VEGF, and VEGFR2, indicating that baicalein suppressed the HIF1α/VEGF cascade in U87 gliomas [112].

Baicalein has been shown to chemically interact with TLR4. Importantly, baicalein binding with TLR4 interrupted the binding of FITC-conjugated LPS and inhibited the activities of TLR4, as evidenced by reduced phosphorylation of NF-κB-p65 and AKT in cells challenged with LPS. Subsequently, inhibition of TLR4 activity remarkably reduced the viability of colorectal cancer cells, which was abolished upon TLR4 overexpression. Furthermore, TLR4 overexpression increased HIF1α and VEGF levels in colorectal cancer cells, while TLR4 knockout led to significant reduction in their levels. TLR4 activation led to an increase in the levels of HIF1α and VEGF, which were reversed by baicalein. Importantly, baicalein substantially reduced NF-κB phosphorylation in the tumor tissues of CRC-bearing xenograft mouse models [113].

The Src pathway is centrally involved in the expression of Id1 (inhibitor of differentiation 1) and promotes carcinogenesis. Intragastrically administered baicalein caused marked reduction in tumor nodules in mice orthotopically injected with A549 cells in the left lung. Essentially, the lungs of tumor-bearing mice demonstrated higher levels of Id1. Baicalein significantly reduced Src phosphorylation in tumor-bearing mice, and these findings highlighted that baicalein inhibited Id1 in an Src-dependent manner [114].

SMYD2 knockdown inhibited proliferation and invasive potential of non-small-cell lung cancer A549 and NCI-H1299 cells. There was an evident shrinkage in the mass of transplanted tumors in mice inoculated with SMYD2 knockdown cancer cells. Moreover, transplanted tumors in SMYD2 knockdown groups were smaller in size. Baicalein significantly inhibited the levels of SMYD2 and RPS7 in NCI-H1299 and A549 cells. Likewise, baicalein inhibited tumor growth and reduced the levels of SMYD2 and RPS7 in the tumor tissues of xenografted mice [115].

TGFβ1 secretion mediated an increase in invasive and metastasizing abilities of MCF7 and MDA-MB-231 cancer cells when co-cultured with M2 macrophages. Essentially, the addition of anti-TGFβ1 neutralizing antibodies before co-culture caused reversal of invasion abilities of MDA-MB-231 and MCF7 cancer cells. Levels of M2-specific markers (CD206) were found to be reduced and levels of M1-specific markers CD86 were increased significantly after baicalein treatment. Data suggested that baicalein changed the phenotypes of macrophages from M2 to M1. Baicalein reduced expression of M2-associated cytokines (TGFβ1, interleukin-10) and enhanced M1-associated cytokines (interleukin-12, TNFα). Co-culture with M2 macrophages significantly upregulated TGFβ1, N-cadherin, and vimentin in MDA-MB-231 and MCF7 cancer cells. After co-culture with M2 macrophages, MDA-MB-231 cancer cells potently promoted tumor growth and pulmonary metastasis. However, baicalein induced phenotypic switching from M2 to M1 and significantly reduced pulmonary metastatic nodules in tumor-bearing mice [116].

## 10. Concluding Remarks

It is exciting to note that cutting-edge research has provided rich and information-dense pictures of protein signaling cascades which play fundamental roles in cancer onset, progression, drug resistance, and loss of apoptosis. Hyperactive/underactive pathways, presence or absence of important feedback mechanisms, and ectopic expression of proteins play central roles in cancer progression. Nevertheless, even this level of understanding may not be sufficient to realistically analyze the therapeutic targets within the context of the cellular networks. Accordingly, it is essential to drill down deep into the intricacies of signaling deregulations to identify signaling molecules and pathways which fuel the survival of cancer cells.

Excitingly, the interplay between inhibition and activation is a universal theme of signal transduction cascades and is mirrored at every level and scale of hierarchically organized protein networks. Therefore, feedback and feedforward loops are the linchpin mechanisms which allow precise and critical control over entire pathways and enable the cells to portray highly complicated modes of response.

In this mini-review, we focused on baicalein-mediated inhibitory effects on oncogenic cell signaling cascades for inhibition of carcinogenesis and metastasis. However, there are wide-ranging mechanisms which are unaddressed mainly in the context of the true potential of the cancer-inhibitory role of baicalein. Future empirical studies must converge on the identification of baicalein-mediated targeting of oncogenic pathways in different cancers. The TGF/SMAD pathway has not been explored in detail to realistically assess how baicalein regulated the TGF/SMAD pathway for cancer chemoprevention. Similarly, SHH/GLI and Hippo pathways have not been comprehensively studied in cell lines and animal models. Likewise, the TRAIL-mediated apoptotic pathway needs further research for clinical trials. Baicalein-mediated upregulation of death receptors and/or activation of the intrinsic apoptotic pathway will be helpful in the design and verification of combinatorial treatments consisting of baicalein and TRAIL-based therapeutics in xenografted mice. Furthermore, baicalein-mediated effects on PDGFR- and VEGFR-induced intracellular signaling need to be tested in different cancers. Another critical and key question regards the identification of target non-coding RNAs of baicalein. LncRNAs and circular RNAs have gained substantial limelight and are reported to be centrally involved in the regulation of various steps of carcinogenesis. Therefore, identification of the most relevant tumor suppressor and oncogenic non-coding RNAs will be useful in combination treatments consisting of baicalein and tumor suppressor mimics or baicalein and oncogenic siRNAs.

An important key question now looms for the field of cancer drug discovery: which proteins can be considered as the most plausible drug targets within a pathway so that result-oriented therapeutic strategies can be rationally designed for proof-of-concept studies regarding the cancer chemopreventive role of baicalein with maximum efficacy and minimum side effects to the patient?

## Figures and Tables

**Figure 1 ijms-23-08377-f001:**
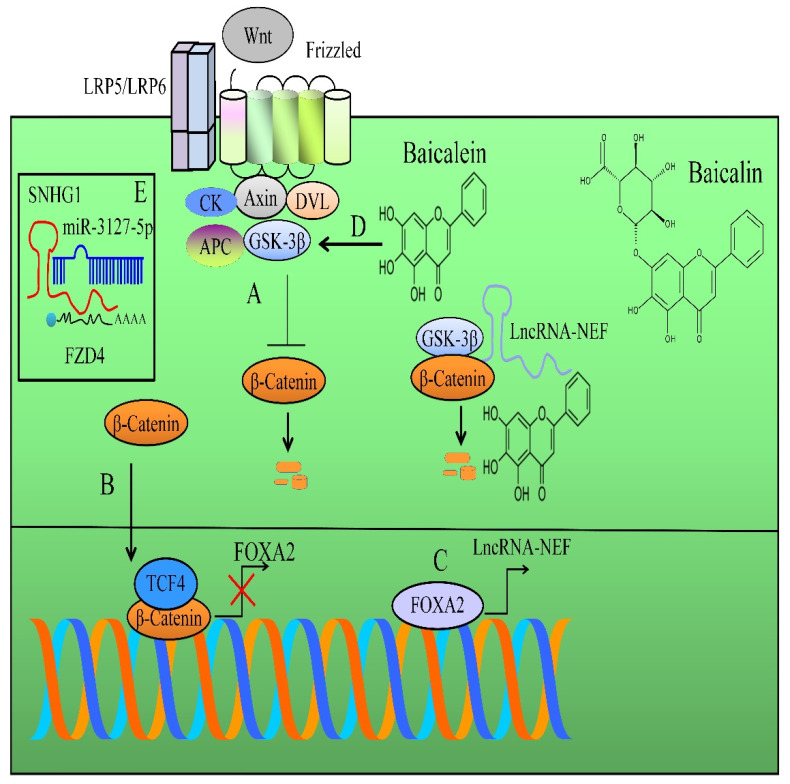
(A) Diagrammatic representation of WNT/β-catenin pathway. APC and axin recruited β-catenin, and after sequential phosphorylations of specified amino acid residues in amino-terminal region, β-catenin is poly-ubiquitinated by β-TrCP and tagged for degradation by the proteasomal machinery. (B,C) β-catenin transcriptionally repressed FOXA2. FOXA2 stimulated the expression of lncRNA-NEF. LncRNA-NEF interacted physically with β-catenin and increased the binding of GSK3β with β-catenin and facilitated phosphorylation and degradation of β-catenin by proteasomal machinery. (D) Baicalein also promoted GSK-3β-mediated phosphorylation of β-catenin and subsequent degradation. (E) SNHG1 antagonized miR-3127-5p-mediated targeting of FZD4.

**Figure 2 ijms-23-08377-f002:**
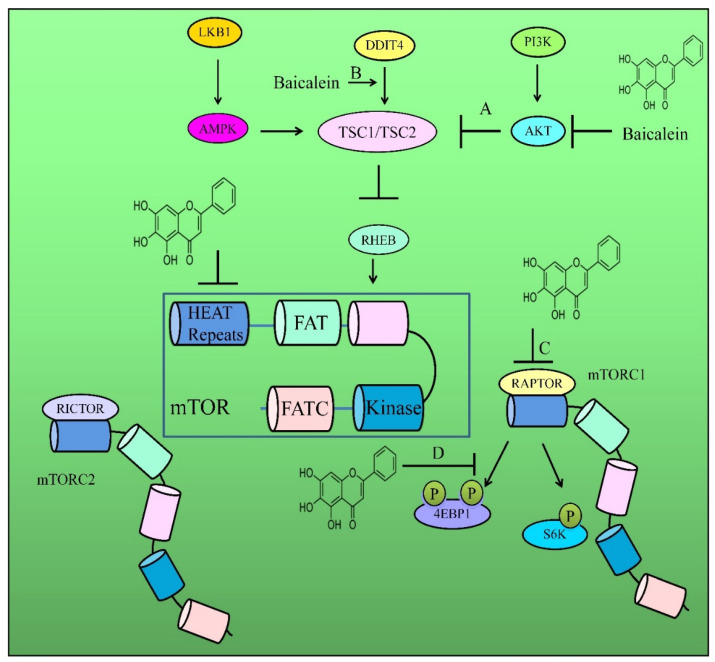
Diagrammatic representation of AKT/mTOR pathway. (A) AKT phosphorylates TSC2 and GTP-loaded RHEB activates mTORC1 (mammalian TOR complex 1). However, TSC1/TSC2-mediated hydrolysis of GTP to GDP switched RHEB from a functionally active GTP-bound state to an inactive GDP-bound state and inhibited the functions of mTOR. (B) DDIT4 has been documented to block the activity of mTORC1 by activation of TSC1/2 complexes. (C,D) Baicalein inhibited mTORC1 (RAPTOR) and mTORC1-mediated phosphorylation of 4EBP1 and S6K.

**Figure 3 ijms-23-08377-f003:**
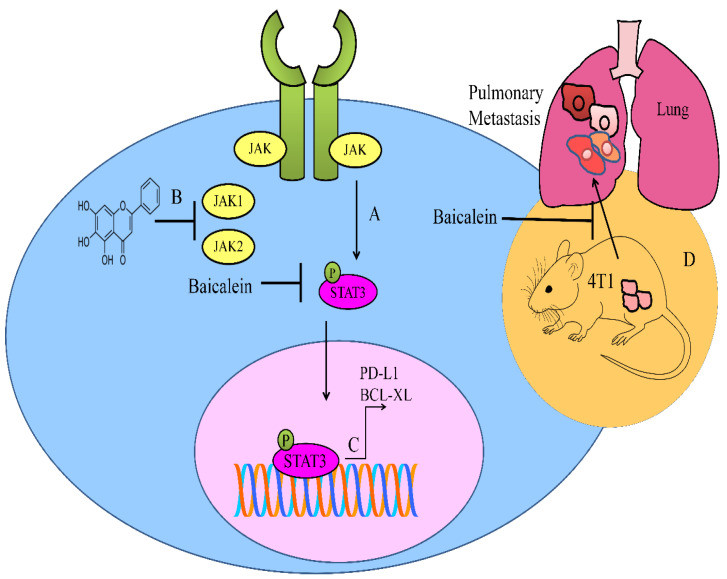
(A–C) Diagrammatic representation of JAK/STAT pathway. STAT3 upregulated PD-L1 and BCL-XL. Baicalein inactivated JAK/STAT pathway. (D) Baicalein inhibited pulmonary metastasis of breast cancer cells in animal model.

**Figure 4 ijms-23-08377-f004:**
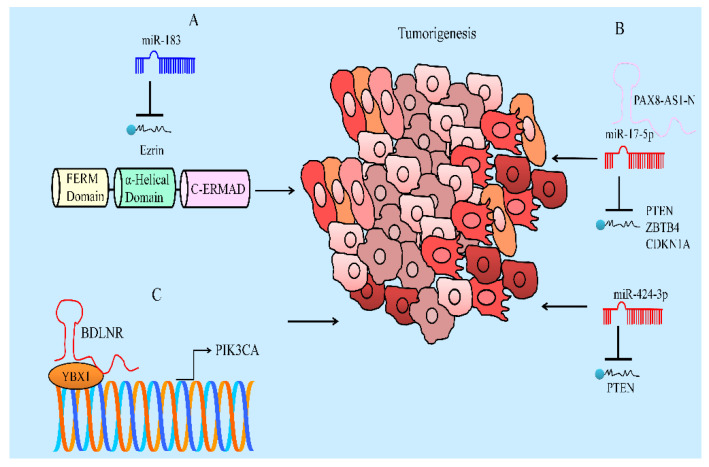
(A) Ezrin is a target gene of tumor suppressor miR-183. miR-183 inhibited tumorigenesis but ezrin promoted tumorigenesis. (B) miR-17-5p and miR-424-3p are oncogenic miRNAs. These miRNAs targeted PTEN. Tumor suppressor lncRNA blocked miR-17-5p-mediated targeting of PTEN and inhibited carcinogenesis. (C) BDLNR physically associated with YBX1 and promoted its binding to PIK3CA promoter and facilitated cancer progression.

**Figure 5 ijms-23-08377-f005:**
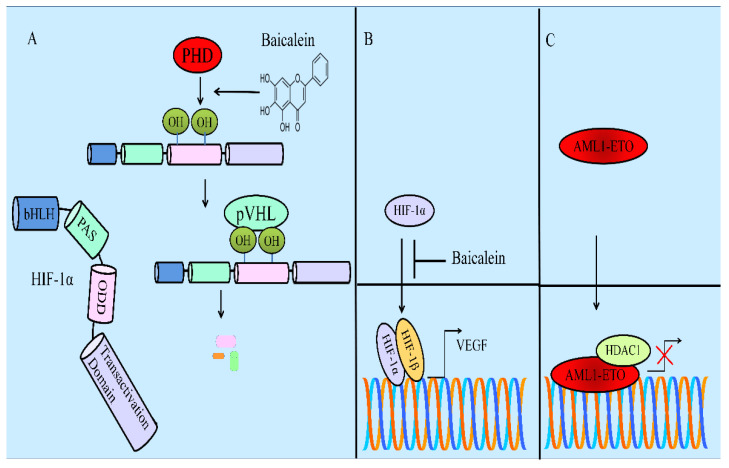
(A) Baicalein induced degradation of HIF-1α by PHD and pVHL. (B) HIF-1α and HIF-1β transcriptionally upregulated VEGF. However, baicalein blocked HIF-1α-mediated VEGF expression. (C) AML1-ETO worked synchronously with HDAC1 and transcriptionally repressed the target genes.

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
