# Peer review of "Regulation of Cell Signaling Pathways and Non-Coding RNAs by Baicalein in Different Cancers"

_ijms, 2022, doi:10.3390/ijms23158377_

Round 1

Reviewer 1 Report

This is an informative review and the authors should be credited for that, however there are some room for improvement on this manuscript. 

1) The authors should add a section in the beginning of the manuscript to describe other biological significance which have been reported for baicalein and its important metabolite, baicalin. Eg. Antiviral, antimicrobial, antioxidant and anti-inflammatory. This will help improve the flow of the review by providing the readers some more background of baicalein, before going into the main details covered under this review.

2) The authors should improve on the style of conveying the different points and look to connect them better. The current review seems to be presented in rigid pieces of facts instead of a good storyline that will enhance the interests of future readers.

Author Response

  • The authors should add a section in the beginning of the manuscript to describe other biological significance which have been reported for baicalein and its important metabolite, baicalin. Eg. Antiviral, antimicrobial, antioxidant and anti-inflammatory. This will help improve the flow of the review by providing the readers some more background of baicalein, before going into the main details covered under this review.

Antibacterial Properties: NorA efflux protein is frequently expressed in multi-drug-resistant (MDR) Staphylococcus aureus strains and is characterized as the main experimental model in the exploration for agents that efficiently inhibit active efflux mechanisms. Baicalein caused a significant reversal of the ciprofloxacin resistance of MRSA by inhibition of the NorA efflux pump (17).

Biofilms formed by Staphylococcus aureus remarkably enhanced resistance against antibiotics by hindering the penetration of antibiotics. Importantly, baicalein inhibited Staphylococcus aureus-induced formation of the biofilms, destroyed bio-films, increased vancomycin permeability, suppressed the production rates of staphylococcal enterotoxin A and α-hemolysin, and inhibited quorum sensing systems (18).

Baicalein reduced the Pseudomonas aeruginosa-induced secretion of the inflammatory cytokines particularly, interleukin-1β, interleukin-6, interleukin-8 and TNFα. Furthermore, baicalein suppressed Pseudomonas aeruginosa-induced activation of the MAPK and NFκB-driven-transduction cascades in co-cultured macrophages (19).

The CTX-M-type β-lactamases confer resistance to expanded-spectrum cephalosporins. Baicalein and cefotaxime synergistically reduced the expression of CTX-M-1 (20).

Serotype 2 (SS2) of Streptococcus suis has been noted to cause severe health com-plications. Baicalein and ampicillin combinatorially reduced inflammation and pathological damages such as high infiltration rates of inflammatory cells, alveolar interstitial congestion, and edema in the brain and lung of mice intraperitoneally infected with SC19 (21).  There are direct pieces of evidence which highlight nanoparticle-mediated delivery of baicalein for effective antibacterial activity (22, 23).

Antiviral Properties: Halogenated baicalein was found to be a promising antiviral agent against SARS-CoV-2 main protease (24). Baicalein and Gallocatechin gallate inhibited the activities of SARS-CoV-2 main protease and blocked replication of the virus. COVID-19 and sepsis have the ability to pathologically trigger cytokine storms. Levels of serum inflammatory factors interleukin-1α, TNFα, interleukin-4, and interleukin-10 in the model group were increased significantly, while they were reduced considerably in mice treated with gallocatechin gallate and baicalein (25). Baicalein inhibited SARS-CoV-2 RNA-dependent-RNA polymerase and exhibited significant antiviral activities (26).

Oxidative stress: Intracerebral hemorrhage (ICH) is a critical and life-threatening subtype of stroke. ICH animal models were established by injection of collagenase into the right of basal ganglia. Baicalein led to an increase in the levels of serum SOD and GSH-Px. Whereas, neuronal apoptosis and pathological injuries of the brain tissues were greatly mitigated. microR-106a-5p directly targeted PHLPP2 but overexpression of PHLPP2 caused reversal of baicalein-mediated effects on ICH mice. Baicalein activated the NRF2/ARE pathway by suppression of PHLPP2 ex-pression (27).

Baicalein enhanced cellular antioxidant defensive capacities through significant reduction in the levels of ROS generation and the activation of the NRF2 transduction cascade, thus protecting C6 cells from H2O2-induced damage of neurons (28).  Co-treatment with H2O2 and baicalein completely suppressed the activation of apoptotic pathway by upregulation of NRF2 expression and reduction in the levels of ROS (29).

Baicalein led to protection of cardiomyocytes against oxidative stress-mediated injuries through NRF2/KEAP1 cascade. Baicalein effectively induced disassembly of NRF2 and KEAP1. Consequently, NRF2 moved from cytoplasm to nucleus and stimulated NRF2/heme oxygenase-1 contents (30).

Baicalein improved the mortality rates, degeneration of neurons, brain water con-tents and cerebral vasospasm in rat models of subarachnoid hemorrhage repeatedly injected with autologous blood. Baicalein also switched-on antioxidant mechanism by activating the functions of SOD and catalase and decreased the levels of malondialdehyde (31).

Baicalein ameliorated myocardial ischemia through reduction of oxidative stress and inflammation (32).

Baicalein improved brain injuries after intracerebral hemorrhage by inhibition of ROS-NLRP3 inflammasomes (33). Overall, baicalein has been demonstrated to improve disease-associated pathological conditions (34-36).

  • The authors should improve on the style of conveying the different points and look to connect them better. The current review seems to be presented in rigid pieces of facts instead of a good storyline that will enhance the interests of future readers.

We have added connecting paragraphs for a better understanding.

Reviewer 2 Report

The abstract is too general or broad about natural products, and the importance of and method of how to collect the data about Baicalein, but it does not mention the main mechanism of action,  e.g. mRNA signaling .
The keyword "non-coding RNA" should be added to the list.

Author Response

Reviewer 2:

The abstract is too general or broad about natural products, and the importance of and method of how to collect the data about Baicalein, but it does not mention the main mechanism of action,  e.g. mRNA signaling

We have edited the abstract and highlighted the edited sections of abstract.

The keyword "non-coding RNA" should be added to the list.

Information has been added in the abstract and in the section of non-coding RNAs.

Reviewer 3 Report

This is an interesting minireview  on the effects of  Baicalein on the regulation of different tumor cell signaling pathways. The manuscript is clear and well organised, and in my opinion can be published as it is. 

Author Response

Reviewer 3:

This is an interesting minireview  on the effects of  Baicalein on the regulation of different tumor cell signaling pathways. The manuscript is clear and well organised, and in my opinion can be published as it is.

Thank you for encouraging comments

Reviewer 4 Report

The present manuscript gives the insight regarding the Regulation of Cell Signaling Pathways and non-coding RNAs 2 by Baicalein in Different Cancers. The review is mainly focused on several signaling pathways related to cancer. The introduction part is short and crispy. The objective of the review is clearly explained. Enough literature were included in the present work. The pictorial representations are clear and self explanatory. I strongly suggest that the authors should provide the structures of baicalin and baicalein separately. The entire manuscript contains lot of abbreviations, but i did not see and list of abbreviations in the manuscript. Hence I strongly advice the authors to include the list of abbreviations in the manuscript. The concluding remarks found to be good. If the authors include the merits and demerits and future directions, it will attract more readers from scientific community.

Author Response

Reviewer 4:

The present manuscript gives the insight regarding the Regulation of Cell Signaling Pathways and non-coding RNAs 2 by Baicalein in Different Cancers. The review is mainly focused on several signaling pathways related to cancer. The introduction part is short and crispy. The objective of the review is clearly explained. Enough literature were included in the present work. The pictorial representations are clear and self explanatory.

I strongly suggest that the authors should provide the structures of baicalin and baicalein separately.

We have formatted and edited the manuscript thoroughly.

The entire manuscript contains lot of abbreviations, but i did not see and list of abbreviations in the manuscript. Hence I strongly advice the authors to include the list of abbreviations in the manuscript.

NLRP3 (NOD-, LRR- and pyrin domain-containing protein 3)

ROS (Reactive oxygen species)

KEAP1 (Kelch-like ECH-associated protein 1)

NRF2 (Nuclear factor erythroid 2-related factor 2)

PHLPP2 (PH Domain And Leucine Rich Repeat Protein Phosphatase 2)

APC (Adenomatous polyposis coli)

CK (Casein kinase)

GSK3 (Glycogen synthase kinase-3)

SCF (SKP1/Cullin/F-box)

β-TrCP (Beta-Transducin Repeat-containing protein)

FZD4 (Frizzled class Receptor 4)

FOXA2 (Forkhead Box A2)

MMP2 (matrix metalloproteinase-2)

SOX2 (SRY-Related HMG-Box Gene 4)

DMH (1,2-dimethylhydrazine dihydrochloride)

DSS (Dextran sodium sulfate)

JAK (Janus Kinase)

STAT (Signal Transducer and Activator of Transcription)

MEK (MAP kinase kinase)

ERK (Extracellular signal-regulated kinase)

AMPK (AMP-activated protein kinase)

RUNX3 (Runt-related transcription factor 3)

PTEN (Phosphatase And Tensin Homolog)

ZBTB4 (Zinc finger and BTB domain containing 4)

The concluding remarks found to be good. If the authors include the merits and demerits and future directions, it will attract more readers from scientific community.

We have improved the concluding remarks
